# Why Vitamin C Could Be an Excellent Complementary Remedy to Conventional Therapies for Breast Cancer

**DOI:** 10.3390/ijms21218397

**Published:** 2020-11-09

**Authors:** Michela Codini

**Affiliations:** Department of Pharmaceutical Science, University of Perugia, 06100 Perugia, Italy; michela.codini@unipg.it; Tel.: +39-075585905

**Keywords:** ascorbic acid, breast cancer, anti-cancer mechanisms, reactive oxygen species, comblementary medicine

## Abstract

The most frequent cancer in women is breast cancer, which is a major cause of death. Currently, there are many pharmacological therapies that have made possible the cure and resolution of this tumor. However, these therapies are accompanied by numerous collateral effects that influence the quality of life (QoL) of the patients to varying degrees. For this reason, attention is turning to the use of complementary medicine to improve QoL. In particular, there are increased trials of intravenous injection of vitamin C at high doses to enhance the antitumor activity of drugs and/or decrease their side effects. This review intends to underline the anticancer mechanisms of vitamin C that could explain its efficacy for treating breast cancer, and why the use of vitamin C at high doses could help patients with breast cancer to enhance the efficacy of pharmacological therapies and/or decrease their side effects.

## 1. Introduction

Worldwide, breast cancer continues to be the most frequent cancer in women. The annual global incidence is over 2 million cases per year, and it causes one of the highest numbers of deaths related to cancer among women as reported by the World Health Organization (WHO) in 2018. In the European Union (EU), breast cancer is the second most frequent cause of death for cancer in women, in spite of the positive trend of mortality from the 1990s [1].

Breast cancer is a multifactorial pathology involving genetic mutations, hormonal disturbances, lifestyle, and family background of breast and ovarian cancer [2]. Having a mother, sister, or daughter (first degree relative) with breast cancer approximately doubles the risk of developing the same type of cancer, and having two first degree relatives increases the risk five-fold. In addition, women with a family background of breast cancer in the male component of the family present an increased risk of breast cancer [2].

Breast cancer is the common term for a set of breast tumor subtypes (most of these are epithelial tumors of ductal or lobular origin), with distinct molecular and cellular origins, defined by distinct pathology, sensitivity to therapy, and molecular characteristics. Breast tumors have been classified into four different subtypes on the basis of the condition of receptors, in particular progesterone receptor (PR), estrogen receptor (ER), and human epidermal growth factor 2 (HER2). The first subtype is luminal A, which is ER and/or PR receptor positive and HER2 negative. The second subtype is luminal B, which is ER and/or PR receptor positive and HER2 positive. The third subtype is *HER2* positive, which is ER/PR negative and HER2 positive; and last one is basal-type tumors, with all the receptors negative [3].

Tumor heterogeneity has become an important clinical feature for diagnosis and therapeutic decision-making. In the EU, the mortality for breast cancer has declined, thanks to early detection and efficient systemic therapies. Recently, there has been an increase in the number of therapies able to save the life of patients affected by breast cancer. Indeed, there are different treatments that can destroy the cancerogenic cells. Unfortunately, each of these treatments is accompanied by a series of collateral effects both in the early and late stages of the disease. These collateral effects are different and depend on the particular drug and its dose and time of treatment [4]. The adoption of complementary medicine to increase the quality of life (QoL) can be considered an important adjunct during treatment. These complementary therapies should be used in women with a lower QoL and with an early stage of cancer [5]. In this case the results of these complementary therapies could give a better QoL to these patients [6]. In this review, the use of vitamin C as a complementary remedy in breast cancers is analyzed.

## 2. Chemistry and Biochemistry of Vitamin C

Ascorbic acid is a water-soluble carbohydrate similar to glucose. However, unlike glucose, it contains the highly reactive “ene-diol” group. This group transforms a relatively inactive sugar to a powerful reducing agent in aqueous solution, which readily donates one or two electrons to radicals and oxidants, generating the relatively stable monodehydroascorbate (MDHA) radical, and the fully oxidized dehydroascorbic acid (DHA). Both DHA and MDHA can be reversibly reduced to ascorbate, as shown in Figure 1 [7,8].

DHA is transported inside the cell by sodium vitamin C transporters 1 and 2 (SVCT1 and SVTC2, respectively) [9]. Inside the cell, DHA can be degraded to 2,3-diketogulonate, oxalate, and *L*-threonate that can be discarded by the kidney [10].

Oxidation of ascorbate by free radicals or reactive oxygen species (ROS) can be performed inside or outside the cell. Therefore, the antioxidant action of ascorbate can decrease the concentration of ROS [11]. On the other hand, when injected intravenously, ascorbate can reach millimolar concentrations, which lead to its action as a pro oxidant [12]. The pro oxidant activity is due to an association with metal ions such as Fe^3+^ and Cu^2+^ that can be chelated by ascorbate [13]. In the presence of oxygen-reduced iron, ions react with hydrogen peroxide (H_2_O_2_) to develop reactive hydroxyl radicals (HO^•^) or peroxide ions (O_2_^•−^) by stimulating the reaction of Fenton (Figure 1) and Haber–Weiss chemistry [14,15].

Vitamin C is an crucial cofactor for many iron- and copper-containing enzymes due to its ability to maintain these transition metals in the reduced state in which the activity of these enzymes is optimized [16].

Vitamin C-dependent enzymes are classified into two groups: copper-containing monooxygenases and iron-containing and α-ketoglutarate-dependent dioxygenases (αKGDDs). αKGDDs uses oxygen and α-ketoglutarate as co-substrates while producing CO_2_ and succinate. Among the reactions catalyzed by αKGDDs are a wide range of hydroxylation reactions such as those involved in collagen synthesis, carnitine synthesis, noradrenaline synthesis, demethylation of protein, DNA and RNA, and hypoxia-inducible factor lα (HIF1α) stability. Thus, vitamin C regulates a variety of fundamental biological processes [13].

In nearly all mammals, ascorbic acid is synthesized in the kidney or liver using glucose from the blood by a number of reactions, as shown in Figure 2. Each reaction, with exclusion of the last one, is regulated by a specific enzyme. In the last reaction, the 2-keto-*L*-gulonolactone after being synthesized is transformed into ascorbic acid.

In humans, mutations dating from millions of years ago have destroyed the ability to synthesize *L*-gulonolactone oxidase, which is an enzyme necessary to transform *L*-gulonolactone into 2-keto-*L*-gulonolactone [12]. This is a clear example of a genetic disease that has previously been considered an avitaminosis.

Since humans are unable to synthesize vitamin C, this is required as an essential dietary supplement [17]. The recommended supplementary dose for an adult is about 100 mg per day in order to generate a 50 micromolar concentration of vitamin C in the plasma. Nonetheless, the concentration of vitamin C is different in different tissues. Circulating leucocytes, pituitary gland, adrenal glands, liver, brain, and skeletal muscle accumulate higher levels of vitamin C than plasma [18]. An elevated concentration of vitamin C in cells seems to indicate the need of ascorbate as a cofactor or to decrease the levels of ROS [19].

## 3. Anticancer Mechanisms of Vitamin C

The increased understanding of the role of ascorbate in the cell together with the molecular mechanisms of cancer development has led to a number of interesting hypotheses regarding the mechanism of vitamin C anti-cancer activity.

The importance of vitamin C in curbing the development of cancer metastasis has been related to its role in collagen synthesis, which is decreased when there is a lack of vitamin C [20]. Indeed, it has been observed that the changes of the stroma surrounding a tumor are identical to those observed in scurvy. Thus, a dense stromal consistency may represent a physical barrier against the spread of neoplastic cells encapsulating them with a dense fibrous tissue. This result can be achieved by high doses of vitamin C. Moreover, vitamin C enhances intercellular ground substance resistance to local infiltration. Hence, it is important to maintain the synthesis of collagen at the right levels by using vitamin C to contrast the development of metastasis [21]. This hypothesis was also supported by the work of Ewan Cameron and Rotman, who suggested that vitamin C inhibits the enzyme hyaluronidase, which decreases the disruption of the tissue and proliferation of cancer cells [22].

It has also been proposed that vitamin C is capable of inhibiting the synthesis of series 2 prostaglandins in cancer cells [23]. These bioactive lipids are known to stimulate cell proliferation, migration, and angiogenesis. Among them, prostaglandin E2 (PGE2) is highly expressed in many solid tumors [24]. In addition, vitamin C counteracts cell proliferation by stabilizing transcription factor protein 53 (P53) [25]. Kim et al. claim that the presence of p53 may represent one of the reasons for differing ascorbate cytotoxicity among cancer cell lines [26].

The structural similarity between vitamin C and glucose facilitates the entry of vitamin C through glucose transporters 1 (GLUT1). Normally, cancer cells have increased expression of GLUT1 transporters due to their increased glucose requirement. The increase in GLUT1 transporters favors the internalization of vitamin C into the cancer cell and promotes its action as a selective, nontoxic drug [12].

In the 2000s, experiments performed in cell culture showed that millimolar vitamin C plasma concentration can kill cancer cells via the pro-oxidative activity of ascorbate, which produces H_2_O_2_ and OH^•^ [27,28].

Iron ions are important for the pro-oxidative activity of vitamin C, and, interestingly, the concentration of labile iron is increased in the microenvironment of tumor cells, which leads to a higher level of labile iron in tumor cells compared to control cells [29].

Furthermore, extracellular H_2_O_2_ can contribute to increase the level of extracellular DHA that after entering the cell can increase the intracellular oxidative microenvironment. DHA is internalized in tumor cells that express high amounts of GLUT1 and generates intracellular oxidative stress due to the reduction of DHA back to ascorbate. This reaction reduces the concentration of antioxidants inside the cells and increases the levels of endogenous ROS causing oxidative cell damage [30].

The anti-tumor effects of vitamin C are not only confined to the stimulation of ROS. It has been suggest that vitamin C enhances the activity of ten–eleven translocation (TET) enzymes through which it can play a role in reprogramming the cells and in the regulation of cell growth. TET enzymes catalyze the oxidation of 5-methylcytosine (5mC) to 5-hydroxymethylcytosine (5hmC), 5-formylcytosine (5fC), and 5-carboxylcytosine (5caC) with the consequence of methylating the DNA and increasing the expression of tumor suppressor genes [31,32]. Vitamin C, as a cofactor, can bind directly to TET enzymes, producing Fe^2+^ from Fe^3+^, which is necessary for TET activity. Compared to other reducing agents able to increase the TET activity, vitamin C is the most efficient one [33].

Therefore, the restoration of TET activity by vitamin C underlines its role in cancer epigenetic reprogramming that includes DNA hypermethylation, which mainly occurs in the promoter regions, and in particular in the CpG island [34].

Another anticancer mechanism of vitamin C occurs through the down-regulation of HIF-1α. Solid tumors are often characterized by low levels of oxygen and high expression levels of hypoxia-inducible factor 1 (HIF1). HIF1 is a transcription factor found in many types of cancers and is composed of HIF-1α and HIF-1β subunits. HIF-1β is a constitutive subunit, whereas the HIF-1α subunit is regulated by O_2_ concentration. O_2_ regulates HIF-1α activity through HIF hydroxylases that require ascorbate as a cofactor. Thus, vitamin C, through the regulation of the HIF-1α subunit, down-regulates HIF1 and the expression of HIF1-dependent genes that are important to activate angiogenesis, glycolysis, resistance to chemotherapy, and the stimulation of a stem cell phenotype, with the consequence of activating metastasis and the growth of a tumor [35,36,37].

Ascorbate can also stimulate the immune system to increase resistance against pathogens. Recently, Wang-Jae Lee reported that ascorbate suppresses the synthesis of interleukin 18 (IL-18), which is a regulator for various type of tumors [38]. The expression of IL-18 is correlated positively with different types of tumors [39]. IL-18 can increase the synthesis of interferon-gamma (IFN-γ) in natural killer (NK) cells, T cells, and activated macrophages [40]. Thus, ascorbate may be efficacious against cancer by decreasing the expression of IL-18, which is able to regulate the escape of different tumor cells, including breast cancer cells, from the immune system.

Finally, yet importantly, vitamin C exhibits anti-inflammatory functions via the modulation of cytokine levels [41]. For example, a concentration of 20 mM ascorbate inhibited the synthesis of interleukin 6 (IL-6) and tumor necrosis factor alpha (TNF-α) in monocytes without changing the levels of interleukin 1 (IL-1) nor interleukin 8 (IL-8) [42]. The same concentration of ascorbate reduced interleukin 2 (IL-2) production in lymphocytes with no changes in the levels of TNF-α and IFN-γ. In cancer patients, ascorbate at millimolar concentrations has been shown to decrease inflammation through suppression of cyclooxygenase 2 (COX-2) and activation of nuclear factor κ-light-chain-enhancer of activated B cells (NF-κB) in endothelial cells [43]. NF-κB is a transcription factor that regulates gene expression in inflammation processes. A low concentration of ascorbate (0.2 mM) increases the expression of NF-κB in Jurkat T-cells [44]. In human cells, ascorbate inhibits TNF-α activation of NF-κB in a dose-dependent manner and can also decrease the synthesis of granulocyte-macrophage colony-stimulating factor (GM-CSF), interleukin 3 (IL-3), and interleukin 5 (IL-5) [45]. It is known that inflammation can regulate angiogenesis, tumor development, tumor proliferation, metastasis, and resistance to therapy [46]. The main vitamin C anticancer mechanisms are summarized in Figure 3.

## 4. Vitamin C Effects in Breast Cancer Cell Lines and Human Mammary Tumor Xenografts

Although data are limited for breast cancer in vivo, in human breast cancer cell lines, vitamin C has been frequently reported to induce apoptosis without having a significant impact on normal cells [47]. Additional data are available regarding the synergistic effect of vitamin C with chemotherapy drugs. Kurbacher et al. investigated the possible synergistic effect between vitamin C and some chemotherapeutics used in breast cancer therapy. They treated two human breast cancer cell lines, MCF-7 and MDA-MB-231, with 1 μM and 100 μM of ascorbate and with the chemotherapics doxorubicin, cisplatin, and paclitaxel. Even if the concentrations of ascorbate were in a normal range, the effect was synergistic with doxorubicin at 1 μM and 100 μM concentrations in MCF-7 and MDA-MB-231. Moreover, in the MDA-MB-231 cells, the effect was dose-dependent. In MCF-7 cells, ascorbate was also synergistic with cisplatin at 1 μM and 100 μM concentrations, while in MDA-MB-231 cells, only 100 μM was effective. Neither concentration of vitamin C improved the activity of paclitaxel in MCF-7 cells, although the lower concentration had a synergistic effect and the higher concentration had an additive effect in MDA-MB-231 cells [48].

Interestingly, Lee et al. showed that a high-dose of vitamin C mediated anti-proliferative effects on various anticancer drug-resistant breast cell lines, including tamoxifen-resistant (TAM-R) MCF-7, long-term estrogen-deprived (LTED) MCF-7, docetaxel-resistant MCF-7, docetaxel-resistant MDA-MB-231, doxorubicin-resistant MCF-7, and doxorubicin-resistant MDA-MB-231 cells. Elevated amounts of vitamin C significantly decreased the cell growth of TAM-R, doxorubicin-resistant MCF-7, and LTED MCF-7, as observed in the MCF-7 cells. Moreover, vitamin C only exerted a slight effect on the normal breast epithelial cells, MCF10A. In addition, anti-proliferative effects were observed at high-doses of vitamin C in doxorubicin-resistant MDA-MB-231 cells and docetaxel-resistant MCF-7 cells, as effectively as in MDAMB-231 cells. Furthermore, the catalase activity of TAM-R MCF-7, LTED MCF-7, docetaxel-resistant MCF7 and MDA-MB-231, and doxorubicin-resistant MCF-7 cells was decreased significantly in comparison to that of MCF10A cells. Therefore, these results indicate that vitamin C at a high dose has a selective anti-proliferative effect on chemotherapy-resistant breast cancer cells. Furthermore, an additive anti-cancer effect when combined with conventional agents was also observed. This provided important evidence that high-dose vitamin C is a promising therapeutic drug, especially when considering that patients with advanced breast cancer ultimately develop resistance to conventional agents [49].

It has been found that an auranofin/vitamin C combination exerted a synergistic and H_2_O_2_-mediated cytotoxicity toward MDA-MB-231 cells and other breast cancer cell lines. The anticancer potential of auranofin/vitamin C combinations was confirmed in vivo using MDA-MB-231 xenografts in mice without notable side effects [50].

Zeng et al. showed that high doses of vitamin C injected intraperitoneally inhibits metastasis of human breast cancer xenografts in nude mice by inhibiting the epithelial–mesenchymal transition [51].

The level of vitamin C at 100 µM in the plasma can be obtained by oral administration. This concentration is able to inhibit triple-negative breast cancer (TNBC) xenograft metastasis in vivo and TNBC cell invasion in vitro [52]. An epigenetic hallmark of breast cancer and other cancers is the loss of 5-hydroxymethylcytosine (5hmC) [53]. Vitamin C at concentration of 100 µM can bring a 5hmC level in TNBC cells similar to those of non-cancerous epithelial breast cells [54].

Although few studies of vitamin C and breast cancer metastasis have been performed in human patients, in vivo animal models support the inhibition of metastasis by administration of vitamin C. For example, vitamin C administrated orally stops the metastasis of murine breast cancer in Gulo knockout mice, which, similarly to humans, cannot synthesize vitamin C [55].

Park S et al. showed that IL-18 promotes the expression of transferrin, which positively regulates cell growth and proliferation in breast cancer cells. This suggest that ascorbate can act against breast cancer, decreasing the expression of IL-18. One of the physiological roles of IL8 is to regulate the escape of various cancer cells, including breast cancer cells, from the immune system [56].

De Francesco et al. reported a synergic effect between vitamin C and dodecyl-tri-phenyl-phophonium (d-TPP) on eradicating breast cancer stem cells (CSCs). In that research, MCF-7 and MB-231 breast cancer cells were treated with d-TPP, an inhibitor of mitochondrial respiration and ATP production. It has been observed that this TPP derivative not only switches the energy pathway towards the glycolytic pathway in these cancer cells but also increases their sensitivity to other metabolic inhibitors like vitamin C and 2-deoxy-*D*-glucose (2-DG) (glycolysis inhibitors), and doxycycline, niclosamide, and berberine (OXPHOS inhibitors). Therefore, that research has shown that vitamin C can present a synergic effect with that of other glycolytic and OX-PHOS inhibitors on the propagation of CSCs. Furthermore, it has been demonstrated that vitamin C is a potential molecule to target the pathway of glycolysis for the effect on CSCs [57]. Tsao C. analyzed the effect in mice of vitamin C on the growth of a human mammary tumor. The addition of ascorbic acid in the drinking water resulted in a significant reduction in the growth of the implanted tumor. The same result was obtained with oral or intraperitoneally administration of a mixture of ascorbic acid and cupric sulfate. No effects were obtained if ascorbic acid and cupric sulfate were administrated individually [58].

## 5. Vitamin C Effects In Vivo Treatment of Cancer in Human

The use of vitamin C for the treatment of cancer was first reported in the 1950s; its involvement in collagen synthesis was the basis to hypothesize that ascorbate replenishment would protect normal tissue to develop cancers and metastasis [59]. In addition, cancer patients show low levels of vitamin C, and administration of vitamin C can improve the immune system [60].

In 1976, Pauling and Cameron published the results of a trial conducted on 100 terminal cancer patients who were given 10 g/day of intravenous vitamin C (IVC) for about 10 days, followed by oral administration of 10 g/day of vitamin C thereafter. A control group of about 1000 cancer patients did not receive vitamin C in any form. Their work showed a 4.2-fold enhanced survival time in the vitamin C treated patients compared with the control group.

Two years later, a retrospective analysis of this study demonstrated that 22% of patients pronounced as terminal patients had a survival time greater than 1 year compared with 0.4% in the control group [4].

Several clinical studies have investigated the effect of vitamin C on QoL in patients affected by breast cancer [61,62].

In a Korean study, IVC therapy significantly ameliorated global QoL scores, with a reduction of nausea and vomiting, fatigue, and an increased appetite [63]. A German study compared patients with breast cancer undergoing IVC therapy with a control group that only received standard therapy. Patients receiving IVC therapy showed an improvement in many clinical symptoms. No collateral effects caused by ascorbate were noticed, with no modifications in tumor status compared to controls [64].

Hoffer et al. described a phase I clinical trial in patients with advanced cancer, receiving up to 1.5 g/kg body weight of IV ascorbate every week, 3 times per week. No adverse effects were observed at any dose, and in two patients the disease was stabilized [65].

Several patients treated with IVC for more than a year have shown substantially reduced grade 1 and 2 side-effects as compared to the control group [66]. Later, Riordan et al. demonstrated that a therapy with 150–710 mg/kg/day ascorbate for up to eight weeks is safe as long as the patient has no history of kidney stone formation [67].

Hoffer et al. described a study in which patients with different cancer types were treated with IVC and chemotherapy. Vitamin C (1.5 g/kg) was administered three times a week during the weeks in which chemotherapy was given, and with at least one day off during the weeks when chemotherapy was not given. Increased energy, functional improvement, and transient stable disease were observed in patients [68].

Some authors have raised fears that vitamin C supplementation might compromise the efficacy of standard therapies because of its antioxidant properties and its accumulation in tumors [69,70]. However, many studies show that, at high concentrations, vitamin C does not interfere with standard therapies and may increase efficacy in some situations [38,71,72,73]. This result is sustained by meta-analyses of clinical studies that show no interference between antioxidant supplementation and efficacy of chemotherapic drugs [74,75].

## 6. Why Should Vitamin C be Used in Breast Cancer Therapy?

Several publications indicate the role of vitamin C as a therapeutic compound in cancer patients, including breast cancer. In particular, the effects of vitamin C treatment were effective in reducing pain, increasing QoL, and increasing appetite [76]. Furthermore, some patients increased their survival time, and vitamin C was an alternative therapeutic strategy for those suffering from chemotherapy. Vitamin C seems to act by reducing oxidative stress, which is one of the most relevant side-effects of chemotherapy and radiotherapy. In fact, the metabolism of tumor cells, radiotherapy, and chemotherapy increase the level of ROS, inducing oxidative stress [64]. The level of vitamin C is related to the stage of the disease; patients with an higher stage of disease showed a lower level of vitamin C, whereas patients with a lower stage of disease showed a higher level of vitamin C [77,78]. An interesting study was performed to evaluate the effects of IVC on patients with primary non-metastasized breast cancer treated with antineoplastic drugs. The efficacy of IVC treatment was equivalent to those obtained with chemotherapy and radiotherapy, but with no side-effects [66]. As reported previously, the patients treated with IVC had a better QoL that could improve the immune system.

A meta-analysis study of vitamin C and survival among women with breast cancer concluded that post-diagnosis therapy with vitamin C may be related to a reduced risk of mortality. Vitamin C administration can significantly reduce the risk of mortality, including the mortality specifically caused by breast cancer [79].

Breast cancer progression is increased by tumor hypoxia that affects angiogenesis, metastatic activity, and cell proliferation [80]. HIF-1α that is responsible for the observed effects is a target of antitumor action of vitamin C. HIF-1-alfa regulates cancer cell metastasis that is the major cause of death for patients with breast cancer [81]. Vitamin C is a cofactor of the enzyme HIF prolylhydroxylase, which is necessary for the degradation of HIF-1α [82].

By studying the molecular mechanisms behind breast cancer development, it has been realized that many of the anticancer mechanisms of vitamin C can be useful for this type of cancer. For example, breast cancer cells are sensitive to hydrogen peroxide generated from pharmacological ascorbate.

As described above, vitamin C regulates the activity of TET enzymes, which is physiologically important to reprogram cells and to control cellular growth. Furthermore, it has been reported that the levels of TET enzymes are significantly reduced in breast cancer, in particular the levels of TET1. In addition, it was demonstrated that 5hmC levels are reduced in several tissues and are related to breast cancers tumorigenesis [83].

It has also been demonstrated in vivo that TET1 inhibits of the progression of breast cancer cells, whereas the reduced expression of TET1 in patients with breast cancer correlates with poor survival [49]. As mentioned above, vitamin C can activate TETs as a cofactor, and it is required for the optimal activity of TETs.

Moreover, *GLUT1* expression is increased in breast cancers with higher grade and proliferative activity, allowing the entry of DHA and its indirect anticancer activity [84].

Finally, vitamin C exerts a powerful anti-inflammatory activity and a immune system stimulation counteracting tumor proliferation, angiogenesis, metastasis, and resistance to therapy [41].

## 7. Conclusions

There are numerous advantages in giving IVC to patients affected by breast cancer, which make it an ideal additional therapy. Cancer patients often have low levels of vitamin C, and IVC provides an efficient strategy to restore the physiological concentrations. Moreover, IVC has been shown to improve QoL in cancer patients. Indeed, both pre-clinical and clinical studies indicate that IVC is able to decrease the toxic side-effects of chemotherapeutic agents without interfering with their anticancer activity, most likely through the antioxidant and anti-inflammatory activities of IVC.

## Figures and Tables

**Figure 1 ijms-21-08397-f001:**
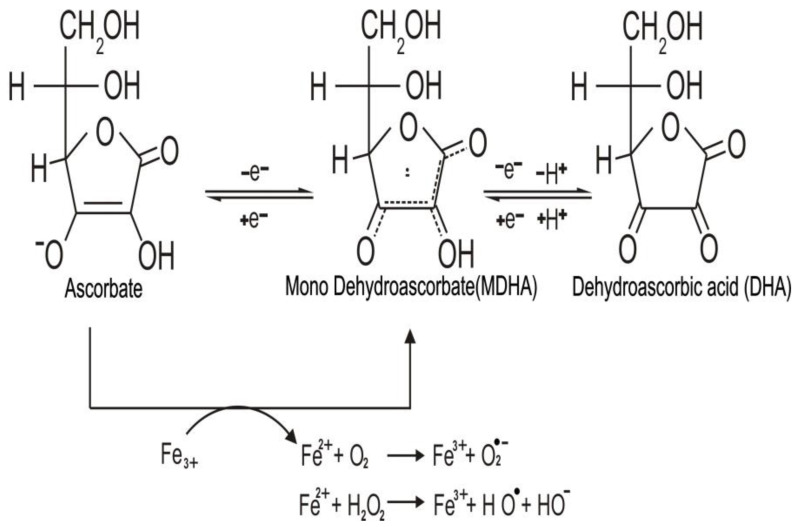
The three redox states of vitamin C (ascorbate, fully reduced form; mono dehydroascorbate, (MDHA), mono-oxidized form; dehydroascorbic acid (DHA), fully oxidized form). Ascorbate can donate one or two electrons to radicals and oxidant agents, forming MDHA and DHA, thus acting as a reducing agent. In the presence of metal ions such as iron, ascorbate can also reduce them and exert pro oxidant effects leading to formation of the superoxide radical (O_2_^•−^), hydrogen peroxide (H_2_O_2_), and hydroxyl radicals (HO^•^).

**Figure 2 ijms-21-08397-f002:**
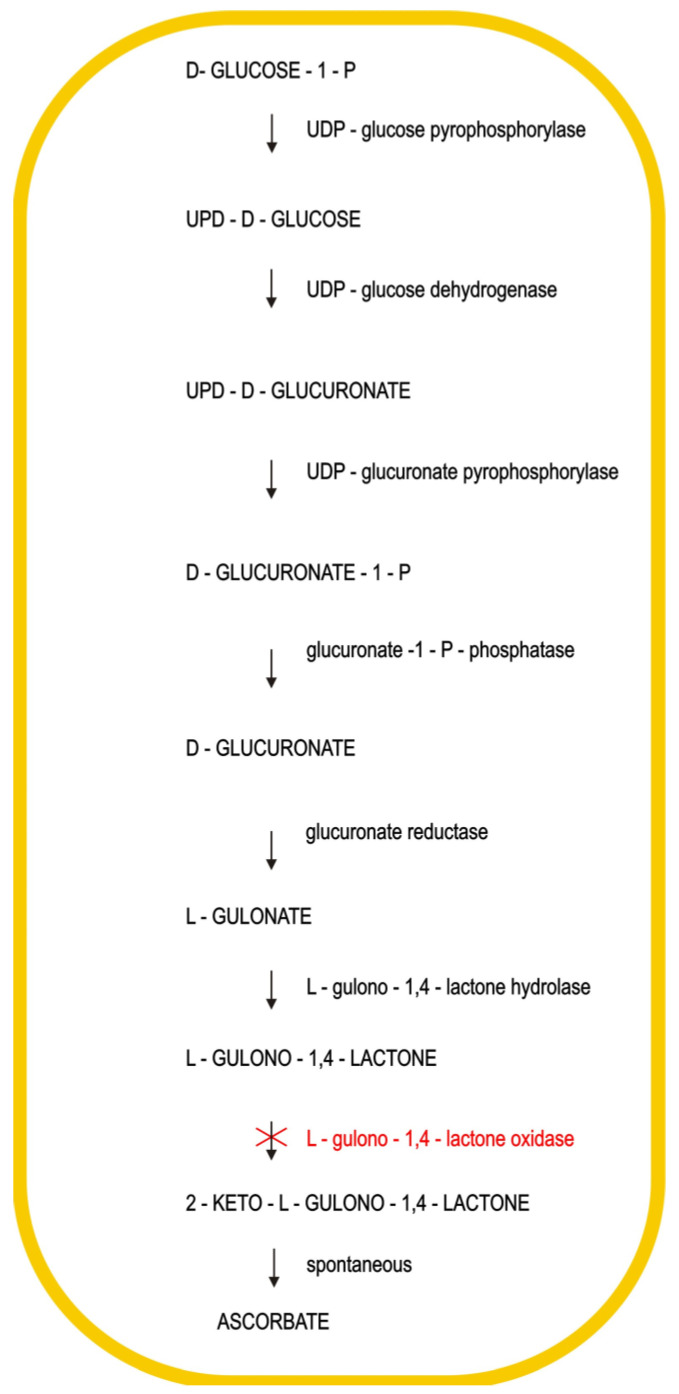
Vitamin C synthesis. The enzyme *L*-gulono-1,4-lactone oxidase is missing in humans and prevents them from producing ascorbic acid from glucose.

**Figure 3 ijms-21-08397-f003:**
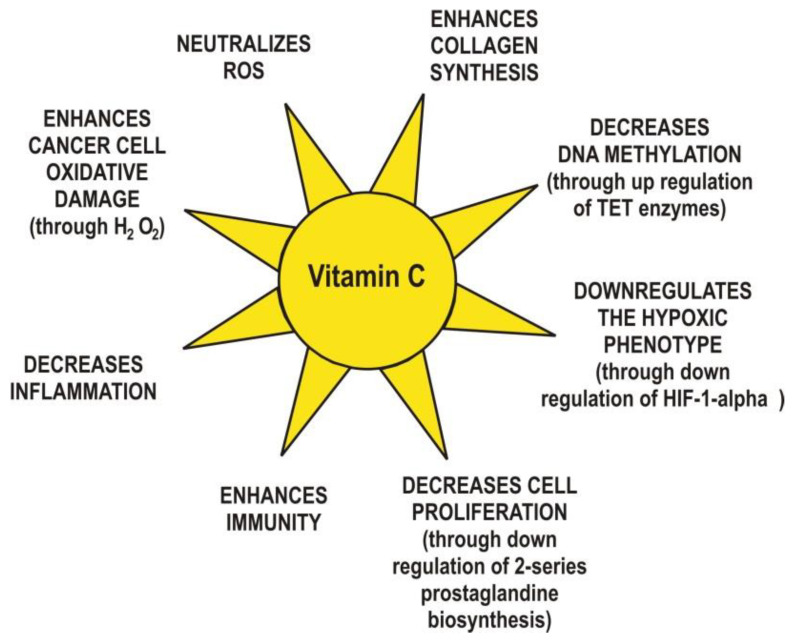
The main anticancer mechanisms proposed for vitamin C.

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
