# Peer review of "Why Vitamin C Could Be an Excellent Complementary Remedy to Conventional Therapies for Breast Cancer"

_ijms, 2020, doi:10.3390/ijms21218397_

Round 1
Reviewer 1 Report
The manuscript named „Why Vitamin C Could Be an Excellent Complemen-tary Remedy to Conventional Therapies for Breast Cancer“ (M. Codini) deals with highly relevant and up-to-date topic.
However, the manuscript needs extensive English editing. In adition, the manuscript lacks a lot of citations, f. e. in the part 3. In the manuscript, there are many abbreviation that are not explained at all. Please, add the list of abbreviations and explain all abbreviations with the first use. In the Figure 3, there are many grammatic errors, please correct it.
In the manuscript, the author describes a lot of molecular mechanisms that vitamin C uses in its anticancer action. Please, add a new figure with the potential anticancer mechanisms of vitamin C in breast cancer.
Author Response
Dear Reviewer, thank you for your comments.
Here are my answers
Best regards
Michela Codini
The manuscript needs estensive English editing.’
The manuscript has undergone an extensive review English editing
‘The manuscript lacks a lot of citations. ‘
Many citations are not present because the policy of the journal is to include primarly citations from 2016 as stated below from the authors guidelines.
Usually, review Paper: A comprehensive Review is recommended, which should contain at least 4000 words text in the main body including 2-3 Figures/Tables. References should be up-to-date, i.e., 50% or above are the papers published within recent 5 years.”
In the manuscript, there are many abbreviation that are not explained at all….please add the list of abbreviations and explain all abbreviations with the first use’
A list of abbreviations has been added and all the abbreviations are explained with the first use.
‘ In figure 3 there are many grammatic errors’
Grammatic errors are now corrected
‘Please, add a new figure with the potential anticancer mechanisms of vitamin C in breast cancer’
As reported before according to Editor requirements only 2-3 figures/tables can be present in the review and therefore the required figure has not been added.
Reviewer 2 Report
This is a timely and well-organized review of the possible roles of ascorbate in breast cancer treatment. Unlike targeted pharmaceuticals, ascorbate has a wide range of interactions and possible effects, making it difficult to discern causal mechanisms of action. The author has performed a credible job of summarizing current thinking regarding the molecule’s chemistry, mechanisms of action, effects on cell lines and xenografts, human studies, and rationale for use. The writing, however, is uneven, and there are numerous grammatical and spelling errors that complicate its interpretation.
- In the Introduction, the author mentions “alternative therapies.” Strictly speaking, alternative medicine is a term that describes medical treatments that are used instead of traditional (mainstream) therapies. Most of the work described in this review considers Vitamin C in a complementary or adjunct setting. The usage of these terms should be clarified.
- Figure 1 and/or its legend needs to be modified to better illustrate how ascorbate donates or accepts electrons to serve alternatively as a reducing or oxidizing agent, and the conditions that drive alternative chemistries.
- The manuscript should more clearly delineate what is known and what is speculative. Given the role of vitamin C in regulation of a variety of fundamental biological processes, the manuscript would benefit from a more systematic treatment of the chemistry, mechanisms, effects, and clinical ramifications. A table linking items in these categories, if possible, and citing evidence would help resolve some of the complexity.
- In the Conclusion, it is not clear what is meant by the phrase, “the same vitamin C exerts an anti-cancer action that can enhance that of the conventional therapies.” The author needs to be clear as to whether existing data support primarily a therapeutic or palliative (addressing quality of life issues) role in cancer treatment regimens.
Author Response
Dear Reviewer, thank you for your comments.
Here are my answers
Best regards
Michela Codini
‘In the Introduction, the author mentions “alternative therapies.” Strictly speaking, alternative medicine is a term that describes medical treatments that are used instead of traditional (mainstream) therapies. Most of the work described in this review considers Vitamin C in a complementary or adjunct setting. The usage of these terms should be clarified.’
The term complementary therapy is now used in the review
‘Figure 1 and/or its legend needs to be modified to better illustrate how ascorbate donates or accepts electrons to serve alternatively as a reducing or oxidizing agent, and the conditions that drive alternative chemistries.’
Figure 1 has been modified according to the referee.
‘The manuscript should more clearly delineate what is known and what is speculative. Given the role of vitamin C in regulation of a variety of fundamental biological processes, the manuscript would benefit from a more systematic treatment of the chemistry, mechanisms, effects, and clinical ramifications. A table linking items in these categories, if possible, and citing evidence would help resolve some of the complexity.’
I agree with your observations but according to Editor requirements (review should contain at least 4000 words text in the main body including 2-3 Figures/Tables) the required table has not been added.
Round 2
Reviewer 1 Report
The manuscript is still not sufficient enough to fullfill the criteria of the journal. The manuscript contains a huge number of stylistic and grammatic mistakes. The sentences at the end often have two dots or they lack it at all. After parentheses stands often no space bar. Please, edit your English and your writing style.
Author Response
Dear Professor,
I had my work revieeed. I hope the manuscript is sufficient enough to fullfil the criteria of the journal.
Best regards
Michela Codini
Reviewer 2 Report
While the content is of interest, the writing and grammar are still substandard and confusing at times. This is evident even in the abstract in phrases such as "the number of patients affected by breast cancer experimenting the intravenous injection of vitamin C at high does, is increasing in order to enhance the antitumor activity of drugs and / or decrease their side effects." At several places within the text, paragraph structure and word choice require changes to ensure reader comprehension.
Author Response
Dear Professor,
I had my work reviewed. I hope the manuscript is sufficient enough to fullfil the criteria of the journal.
Best regards
Michela Codini
Round 3
Reviewer 1 Report
Even after the recommendations the manuscript contains the same mistakes as before. In the line 137, the space bar is missing (OH•[27,28] instead of OH• [27,28]).
In the line 179 ...suppression of cyclooxygenase 2 (cox-2) and activation of nuclear factor kappa.... Is in that case COX-2 the enzyme? Or what exactly do you mean? If COX-2 is already enzyme, it should be written with capital font.
The latin words „in vitro“ and „in vivo“ should be written in italics.
Line 322, after the citation there are two dots. Please correct.
Line 327 contains the information about glut1. If it is already the protein, please, write it in capital font!
The line 330 has no citation. Please add
Author Response
Dear Professor,
thank you very much for your comments.
In this revised version of the manuscript I have included all your observations.
My best regards
Michela Codini
Reviewer 2 Report
No further concerns
Author Response
Thank you for reviewing.
This manuscript is a resubmission of an earlier submission. The following is a list of the peer review reports and author responses from that submission.